# Factors of Prison Recidivism in Women: A Socioeducational and Sustainable Development Analysis

Elisabet Moles-López [1] and Fanny T. Añaños [1,2,*]

1 Department of Pedagogy, University of Granada, 18011 Granada, Spain; elisabethmoles@ugr.es
2 Institute of Peace and Conflicts (IPAZ), University of Granada, 18011 Granada, Spain
* Correspondence: fanntab@ugr.es

**Abstract:** This article analyses women sentenced to prison in Spain (in open, closed, or ordinary regimes) to examine the risk and protection factors fundamentally related to criminal-prison recidivism. The study is national in scope, with a stratified representative sample of 756 female inmates, 446 in a closed environment and 310 in an open one. The women were surveyed using mixed questionnaires, and the data were analyzed using various tests and quantitative models of analysis. The results show the relationship between recidivism and different sociological and criminological characteristics. They highlight national origin as a risk factor for both samples. Age, addiction profile, criminal antecedents as a minor, and age at which the woman first went to prison were also assessed. Protective factors included official education, with education during the sentence as the most important, as well as courses and/or intervention programs attended. All of these issues contribute not only basic knowledge on the topic—for women while in prison and on temporary release—but also the foundations for improving prison socioeducational interventions/treatments and sustainable human development, and for preventing recidivism based on the best, most effective programs that focus on these issues.

**Keywords:** recidivism; prison; risk factors; protective factors; education; gender; reintegration; sustainable development goals (SDGs)

## 1. Introduction

Article 25.2 of the Spanish Constitution states that "punishments that deprive people of liberty and security measures are oriented to re-education and social reintegration". Reintegration is understood as a process of recovery, re-education, and socialization that involves the deployment of multiple possibilities and resources to enable the individual to pursue a successful process of integration–reintegration into the community to which she belongs and thus to prevent recidivism. It is in this context that various intervention programs focusing on different areas gain importance. Among these, we would highlight educational and training programs delivered in the prison environment, an environment also configured as a social–cultural space that focuses on preparing inmates for their freedom. The prison space thus influences the development of community preventive functions, while also generating what Añaños [1] termed new proposals through elements and actions oriented to socialization and prevention of recidivism.

These foundations are endorsed by the United Nations Office on Drugs and Crime (UNODC) [2], which identifies crime prevention as closely related to development and sustainable living environments. The latter is the most important in countries with low and medium-level incomes. Further, the focus on human development promoted by the sustainable development goals (SDGs) [3] concentrates on people to develop their full potential and thus increase their opportunities and choices, encouraging high-quality education, equality of women, access to justice, reduction of inequalities, social inclusion, health, and wellbeing, among other issues. These principles also form part of Human

Rights. Although a person is sentenced to prison, loss of liberty does not exempt that person from protection and enjoyment of those rights [4].

The repercussions of these aspects of human development are of great importance. Data from the Spanish Secretary General of Prisons (hereafter, SGIP) [5] numbers the inmate population in Spain as 58,369 persons in January 2020, of whom 92.6% were men and 7.4% women. Another very important issue is the increase in the population of women prisoners, a worldwide phenomenon since 2000. The female prison population has grown much more rapidly than the male population, at a ratio of 53.3 to 24% [6], although the number of women is still far lower than the number of men in the total prison population.

It is also important to define recidivism as used in this study. The concept refers to repeated criminal behavior [7,8], linked to successful social reintegration processes.

The recidivism rate in Spain is approximately 31.6% [9] in general and around 24.8% [10] for women. Given its importance, it is necessary to explain the perspective we adopt in this study, since the term has different properties, especially of a legal nature. We can distinguish different types of recidivism: guilty pleas, criminal, police, penal, judicial, prison-related, and legal [7,11]. This study will base its analysis on what is termed prison recidivism, which involves returning to prison for having committed a new crime. The new crime may or may not be the same type of crime as the previous one, but it must have occurred at a date after definitive release from prison [9].

Few studies have been performed on recidivism, but it is taken into account when analyzing the evolution of degrees or levels of prison treatment in inmates (both male and female) and thus defines the conditions and contexts for serving a sentence. Because these studies have hardly considered gender, however [10], it is significant in this context to determine the characteristics and differences in gender, as women have different ways of being, attitudes, emotions, and conceptions about the topic [12–18].

Factors influencing recidivism are related to the confluence of diverse factors: (a) risk factors (which increase the risk of recidivism) and (b) protective factors (which decrease this probability significantly) [19].

We can classify risk factors of recidivism into two categories [20]:

(a) Static factors: factors that cannot be changed and that are important in evaluating the risk of recidivism. Following Graña, Andreu, and Silva [21], and based on an exhaustive review of other authors, static factors are:

- The age at which the crime was committed and at which the first arrest occurred. Minors are those most likely to commit another offense [22,23].
- Criminal history—juvenile and adult [21,22,24,25].
- Crime in the family.
- Family relationships, according to parental socialization and its relationship to social anxiety and victimization [26].
- Education level [23,27].
- Family structure (children, partner).
- Nationality—foreign or Spanish [28].
- Socioeconomic status [29,30].

(b) Dynamic factors: factors that can change and that are important not only in assessing risk but also in designing the different intervention programs in the prison environment. These types of factors include:

- Antisocial personality, or mental or personality disorders [27,31–33].
- Substance abuse [23,34–40].
- Type of crime.

Protective factors are associated with the specific person's capability for resilience [41,42]. Resilience becomes very important for the three types of prevention—primary, secondary, and tertiary [36]. Primary—or universal—prevention anticipates the risk of antisocial behavior and faces it before it appears. Secondary—or selective—prevention is both individual and collective in character. Tertiary—or indicated—prevention is directed to

individuals who have already committed a crime and seeks to make them aware of the damage caused [42].

Protective factors [2], in turn, are defined as:

– Factors related to human capital: that is, those that include the person's capability to make changes and achieve goals. Here, social and intellectual skills enter into play [43].

– Factors related to social capital: for example, job or family, through support for and promotion of good behavior [24,44,45].

Based on the preceding, the main aim of this study is to analyze women inmates in both ordinary, closed, and open environments and the factors (both risk and protective) that influence recidivism in Spain in order to improve the prevention of recidivism. We tackle this problem from an approach involving socioeducation, the SGDs, and gender.

## 2. Materials and Methods

This study's methodological approach is based on two research projects performed in Spain: In MUDRES (Drug-dependent women inmates and their social reintegration: A socio-educational study and proposals for action) (Ref. EDU2009-13408), which focuses on closed or ordinary prison environments and was carried out during 2010–2013, 42 prisons were visited in 11 autonomous communities. In REINAC (Reintegration and support processes for women on temporary release) (Ref. EDU2016-79322-R), carried out 2017–2020, 31 prisons were visited in 13 autonomous communities.

The study was approved by the Ethics Commission of the General Deputy Director of Institutional Relations and Territorial Coordination, the General Secretariat of Prisons, and the Department of Justice of the Government of Catalonia. It was also governed by the main ethical principles for studies and research with human subjects in effect at the University of Granada. In all cases, the study was performed with the formal written and voluntary consent of the women composing the sample.

The total sample includes women in two living regimes in the prison environment (MUDRES: closed or ordinary environment and REINAC: open environment or temporary release). In both cases, stratification was performed with allocation proportional to the women's prison population by geographic area. This process obtained highly significant values for the total Spanish prison population. We used two-stage sampling to obtain the sample. First, as indicated above, we chose the prisons by regional representation and ratio of women. Second, we chose at random from among the women who wished to participate. The selection criteria have been fundamentally, in the case of women who were open or temporarily or "semi-liberty" in solitary confinement, the degree of conviction of all women, in 3rd grade and 2nd special degree, who had previously been in prison.

We should clarify that this is not a longitudinal study. Rather, it uses two R&D&I research projects, performed at different times with different populations, that have a common goal—to analyze women's recidivism. The difference in coverage between the two studies is also due to the number of prisons with an open regime or temporary release. The national sample of women inmates in this context is lower than the number of prisons and women in the closed or ordinary context.

The women in the MUDRES study were serving sentences classified as second-degree. The population from which the project's sample was drawn was 3484 women, surveyed from June to October 2011. The number of valid questionnaires was 446 (margin of sampling error $\pm$ 3.9 points), which represents 17% of the total population of women in prison in Spain. In the REINAC study, the sample was composed of women in an open environment who were serving third- or special second-degree sentences, within the framework of a total population of 1062 women throughout Spain. This survey produced a sample of 310 questionnaires (margin of sampling error $\pm$ 4.5 points), representing 30.1%. The fieldwork was performed from June 2018 to March 2019.

Table 1 shows that 71.2% of the total sample did not become recidivists, while 28.8% did. Of the sample, in the closed environment (MUDRES), 75% became recidivists, and

25% did not. In the open environment (REINAC), only 24.8% became recidivists, and 75.2% did not.

**Table 1.** Definition of recidivism as the main sample.

|  | TOTAL | MUDRES | REINAC |
|---|---|---|---|
| Recidivist | 28.8% (218) | 31.6% (141) | 24.8% (77) |
| Non-recidivist | 71.2% (538) | 68.4% (305) | 75.2% (233) |

Source: The authors (Projects Ref. EDU2009-13408 and Ref. EDU2016-79322-R).

The average age of the population was 38.52 years for the total sample, 36.41 for MUDRES, and 42.19 for REINAC.

By national origin, the percentage of foreign women was 25% in the total sample, 26.2% in MUDRES, and 22% in REINAC. Most of the foreign women come from Latin America, and the others are from the European Union.

As to marital status, in the total sample, 18.4% were married, 17.8% living together, 39.2% single, 20.2% separated/divorced, and 4.4% widowed. In MUDRES, the percentages were: 17% married, 20% living together, 38.2% single, 18.9% separated/divorced, and 5.9% widowed. In REINAC, they were: 20.5% married, 14.6% living together, 40.6% single, 22.1% separated/divorced, and 2.3% widowed.

As to drug addiction profile at the time of the study, the highest percentage in all cases—the total sample as well as MUDRES and REINAC—was non-Addicts (NA), 53.4%, 37.2%, and 76.8%, respectively. The next-most-frequent profile was ex-addict (EX), 26.2%, 35.9%, and 12.3%; followed by methadone maintenance treatment (MMT), 11%, 13.7%, and 7.1%. Finally, active addicts (AA) were 9.4%, 13.2%, and 3.9%.

### 2.1. Instruments

The data collection instruments were prepared ad hoc. In both cases, the content was evaluated by external judges and a pilot study to validate the instruments. The MUDRES study used a 92-item questionnaire divided into four main sections and the REINAC study a questionnaire of 115 items in six main sections. Both studies asked for sociodemographic and legal information and included questions about socioeconomic issues, social inclusion, education and programs, prisonization and support relationships, health and drugs, and gender violence and conflicts. The questionnaires were administered in person or in small groups and could be completed by the women themselves, with guidance from others, or through a mix of self-completion and guidance, based on each woman's characteristics.

### 2.2. Data Analysis

The information was analyzed using quantitative methods following the design of a database in SPSS versions 23 and 24 (Statistical Package for the Social Sciences). The analyses performed were bi-variate descriptive. Calculation of Pearson's Chi-square test enabled identification of the presence or absence of relationships among various variables of interest. The relationships among the variables were then used to propose binary logistical regression models. To confirm the models' validity and subsequently to draw conclusions, various tests of validity and fit were performed, including Nagelkerke's R-squared, Pearson's Chi-square test, and the Hosmer–Lemeshow test.

The specific variables analyzed were:

- The risk factors of age, national origin, education level, marital status, children, income level, intimate partner abuse, self-perceived symptomatology of undiagnosed mental illness (disorders such as depressive anxiety; bipolar, personality, and eating disorders; and schizophrenia), drug profile before prison and at time of the study (based on the classification developed by Añaños [46]: active addict (AA), non-addict (NA), in methadone maintenance treatment (MMT), and ex-addict (EX)), criminal antecedents

such as a minor, age the respondent first went to prison, having people close to the respondent who were in prison, and type of crime.

— The protective factors of a person(s) with whom the respondent lived before prison, person(s) with whom she would live after the sentence, employment status before entering prison, courses and/or programs attended during the sentence, official education during the sentence, family support before prison, support from friends before prison, family support now, and support from friends now.

## 3. Results

The following results correspond to a set of questions on the questionnaire.

### 3.1. Relationship to Sociodemographic Variables

Table 2 shows a relationship between *age* and whether or not the person committed another offense in both the total sample and the closed environment (MUDRES), with a *p*-value of 0.000 (less than 0.05). The Chi-square indicates a relationship between the two variables. In the REINAC study, in contrast, no relationship is found.

**Table 2.** Variables for risk factors of recidivism, according to significant results of the $\chi^2$ test.

|  | **TOTAL** | **MUDRES** | **REINAC** |
|---|---|---|---|
| Age | 0.000 *** | 0.000 *** | 0.158 |
| National origin | 0.000 *** | 0.000 *** | 0.000 *** |
| Education level | 0.000 *** | 0.000 *** | 0.000 *** |
| Marital status | 0.177 | 0.260 | 0.003 *** |
| Children | 0.112 | 0.575 | 0.037 ** |
| Income level | 0.684 | 0.996 | 0.167 |
| Intimate partner abuse | 0.507 | 0.034 ** | 0.177 |
| Mental health problems | 0.005 *** | 0.135 | 0.013 *** |
| Drug profile before prison | 0.000 *** | 0.000 *** | 0.090 |
| Current drug profile (last 30 days) | 0.000 *** | 0.000 *** | 0.056 |
| Antecedents as a minor | 0.019 *** | 0.654 | 0.000 *** |
| Age first went to prison | 0.000 *** | 0.716 | 0.000 *** |
| People close to respondent who are in prison | 0.000 *** | 0.000 *** | 0.001 *** |
| Type of crime | 0.000 *** | 0.000 *** | 0.006 *** |

*** Significant at 99%; ** significant at 95%. Source: The authors: (Projects Ref. EDU2009-13408 and Ref. EDU2016-79322-R).

For the total sample, the most significant percentage distributions of these relationships were age group, with the most recidivists: 36–45 years old, with 48.8% (106), followed by ages 26–35 with 27.6% (60), and, to a lesser extent, women over 61 and 18–25. The distribution is similar for non-recidivists.

In the closed environment, the age group with the most recidivists was 36–45 years old, with 52.1% (73). The age groups with the fewest recidivists were 61 or over, with 0.7% (1) and 18–25, with 1.4% (2). The most common age group among non-recidivists was 26–45, with 36.3% (110). The least common groups (lowest percentage) are those of older women, 61 or older, and 48–60.

National origin is another variable related to whether or not a respondent commits another crime. In all three samples (total sample, MUDRES, and REINAC), most of the women recidivists were Spanish (75%, 72.9%, and 78%, respectively). In the three samples chosen, a *p*-value = 0.000 < 0.05 indicates a significant relationship among the variables. In the total sample, 94% (203) of the recidivists were Spanish and 6% (13) foreign. Following

the same line, MUDRES and REINAC showed that 92.9% (131) and 96% (72) of recidivists were Spanish and 7.1% (10) and 4% (3) foreign, respectively, in the two studies.

The significance of the variable education level also implied a relationship to recidivism, *p*-value = 0.000 < 0.05, in all three samples. For the total sample, the education levels of women with the highest percentages of recidivism were primary 45.5% (97), no education 14.5% (35), and vocational training (VT) 12.1% (26). The education levels with the lowest percentage of recidivism were non-compulsory secondary education (Bachillerato/COU) and university study. The education levels with the highest percentage for non-recidivists were primary education 29.7% (158) and secondary education 28.6%. The lowest percentage was for women with no education, 6.4% (34).

In MUDRES, the majority of recidivists had primary education, with 45.3% (62), vocational training (VT), with 16.8% (23), and no education 10.9% (15). The education levels with the lowest percentage of recidivists were university study, incomplete secondary education, and Bachillerato/COU. Among non-recidivists, 33.2% (100) had secondary education, 29.9% (90), 13% FP, and 9% higher education. Only 5.3% had no education.

For REINAC, the education level for the greatest percentage of recidivists was primary education 45.5% (45), followed by no education 20.6% (16). The education levels with the lowest percentage were Bachillerato/COU, vocational training, and higher education. Among non-recidivists, 29.5% (68) had primary education, 22.5% (52) secondary education, 16% (37) Bachillerato/COU, and 12.6% (29) higher education. Only 7.8% (18) had no education.

In REINAC, the results of the chi-square test also indicated a relationship of recidivism to marital status, *p*-value = 0.003 < 0.005, and to children, *p*-value = 0.037 < 0.005. No relationship was established in the total sample or in MUDRES.

The distribution of recidivists by marital status was 70.2% (54) without a partner (single, divorced-separated, or widowed), 20.8% (16) married, 9.1% (7) in domestic partnerships, and 7.8% (6) widowed. Among non-recidivists, 63.2% (146) had no partner, 20.3% (47) were married, and 16.5% were in domestic partnerships.

As to the relationship between recidivism and children, 90.9% (70) of the recidivists had children and 9.1% (7) did not. Among non-recidivists, 80.7% (188) had children and 19.3% (45) did not.

### 3.2. Relationship to Gender Violence and Mental Health

The chi-square tests indicated a relationship between recidivism and gender violence in the MUDRES sample (*p*-value = 0.034 < 0.005). Of recidivists, 57.5% (77) had suffered abuse and 42.5% (57) had not. Of non-recidivists, 68% (200) had been abused, and 32% (94) had not.

The study also found a relationship between recidivism and mental health, in both the total sample (*p*-value = 0.006) and REINAC (*p*-value = 0.013). In the total sample, 85.2% (196) of recidivists showed some symptomatology of formally undiagnosed mental illness, and 14.8% (34) did not. In REINAC, 86.8% (66) of recidivists had some symptomatology, and 13.2% (10) did not.

### 3.3. Relationship to Addiction Profile

This variable included information from two different times, before going to prison and at the time of the study. In both the total sample and in MUDRES, a relationship was found between addiction profile (as defined in the classification used in the MUDRES project (Añaños, 2017)) (at both times) and whether or not the respondent was a recidivist (*p*-value = 0.000 < 0.005). However, no relationship was found in REINAC.

The percentage distribution for the addiction profile before prison for the total sample showed that 64.7% (141) of women recidivists were AA, 27.5% (60) NA, 5.5% (12) in MMT, and 2.3% (5) EX. In MUDRES, 78.7% (111) of the women recidivists were AA, 10.6% (15) NA, 7.1% (10) in PMM, and 3.5% (5) EX.

As to current addiction profile (past 30 days), 49.5% (108) of women recidivists in the total sample were NA, 27.5% (60) PMM, 11.5% (25) AA, and 11.5% (25) EX. In MUDRES,

44.7% (63) of women recidivists were NA, 29.2% (42) in PMM, 14.9% (21) AA, and 10.6% (15) EX.

### 3.4. Relationship to Criminal Variables

Analysis of the relationship of criminal variables to recidivism included the variables antecedents as a minor, age the woman first went to prison, having people close to one who were in prison, and type of crime.

First, a relationship was observed between recidivism and antecedents as a minor in the total sample and in the open environment (REINAC), with *p*-values = 0.019 and 0.000, respectively. In the total sample, 13.8% (30) of women recidivists had criminal antecedents as minors, and 86.2% (187) did not. In REINAC, 10.4% (8) of women recidivists had criminal antecedents as minors, and 89.6% (69) did not.

Second, the significance of the $\chi^2$ test indicated a positive relationship between recidivism and age the respondent first went to prison. As in the previous case, this relationship was found for the total sample and for REINAC, with a *p*-value = 0.000 in both cases. For the total sample, the age group first entering prison with the highest percentage of recidivists was 18–25 (42.4% (89)), followed by 26–36 (37.6% (79)). The least common age groups were over 50 (0.5% (1)) and 37–49 (8.1% (17)). In REINAC, the age groups when first entering prison that showed the most recidivists were 18–25 and 26–36 (42.7% (32)). The age groups with the fewest recidivists were over 50 (1.3% (1)) and 37–49 (13.3% (10)).

Third, a relationship was established to the variable persons close to respondent who were in prison in the total sample, MUDRES (*p*-value = 0.000 in both cases), and REINAC (*p*-value = 0.001). In the total sample, the percentage distribution for recidivists was 66% (142) who had persons close to them who were in prison and 34% (73) who did not. For MUDRES, 68.3% (95) recidivists had someone close to them who was in prison, and 31.7% (44) did not. In REINAC, 61.8% (47) had someone close to them who was in prison, and 38.7% (29) did not.

Finally, as with the previous variable, a relationship was found between recidivism and type of crime in the total sample, MUDRES (*p*-value = 0.000 in both cases), and REINAC (*p*-value = 0.001). Among recidivists in the total sample, the most common crime was against property and the socioeconomic order (56% (116)), followed by crimes against collective security (33.3% (69)) and the integrity of persons (5.8% (12)). For MUDRES, the most common type of crime among recidivists was against property and the socioeconomic order (56.8% (75)), followed by collective security 31.1% (41), and finally, the integrity of persons (4.5% (6)). In REINAC, the results from most to least common type of crime were against property and the socioeconomic order (54.7% (41)), collective security (37.3% (28)), and finally, the integrity of persons (8% (6)).

### 3.5. Models of Risk Factors Proposed

In Table 3, the models proposed show good goodness of fit, with the model corresponding to the whole sample (TOTAL) showing the best goodness of fit. In all cases, we see that $R^2$ ranges from 0 to 1 (*p*-value = 0.749 for TOTAL, *p*-value = 0.432 for MUDRES, and *p*-value = 0.380 for REINAC). These results indicated that the variables introduced were sufficient to explain the differences between becoming a recidivist (yes or no), with the $R^2$ of the model for the total sample taking the highest value.

**Table 3.** Tests for validity and fit of the models proposed for risk factors.

|  | TOTAL | MUDRES | REINAC |
|---|---|---|---|
| Nagelkerke's R-squared | 0.749 | 0.430 | 0.487 |
| $\chi^2$ Test | 0.000 | 0.000 | 0.000 |
| Hosmer-Lemeshow Test | 0.123 | 0.771 | 0.449 |

Source: The authors (Projects Ref. EDU2009-13408 and Ref. EDU2016-79322-R).

For the $\chi^2$ test, which determines whether the model is globally significant, both models showed *p*-values equal to 0, indicating that both models were valid according to this test.

When the Hosmer–Lemeshow is performed to evaluate the model's goodness and solidity, however, all cases obtain values above 0.05 (*p*-value = 0.123 for TOTAL, *p*-value = 0.123 for MUDRES, and *p*-value = 0.885 for REINAC). These results, along with the $\chi^2$ test, indicated that the models were valid for drawing conclusions even though $R^2$ was small.

Table 4 displays the results obtained in the models proposed in this study, one for the closed environment (MUDRES) and one for the open environment (REINAC), to analyze the relationship between recidivism and different variables very important to possible risk factors identified in the theoretical framework.

**Table 4.** Models proposed on the relationship of recidivism profile to significant variables for risk factors.

|  | TOTAL | MUDRES | REINAC |
|---|---|---|---|
| **Age:** |  |  |  |
| 18–25 years | −6.930 *** | −2.521 | − |
| 26–35 years | −4.318 *** | −0.332 | − |
| 36–45 years | −2.420 ** | 0.410 | − |
| 46–60 years | −1.491 | 0.630 | − |
| **National origin:** |  |  |  |
| Spanish | 2.003 *** | 0.922 ** | 2.404 *** |
| **Education level:** |  |  |  |
| No education | 0.875 | 0.716 | 1.063 |
| Incomplete primary | 0.684 | 0.996 | 0.607 |
| Complete primary | 0.391 | 0.946 | 0.268 |
| Incomplete secondary (ESO/BUP) | 0.288 | 0.328 | −0.364 |
| Complete secondary (ESO/BUP) | −0.133 | 0.200 | −0.582 |
| Official non-university vocational training (VT) | −0.649 | 0.378 | −0.964 |
| University higher education | 0.380 | 0.437 | 1.063 |
| **Marital status:** |  |  |  |
| Married | − | − | −0.786 |
| Domestic partner | − | − | −1.612 ** |
| Single | − | − | −0.820 |
| Children | − | − | 0.512 |
| Gender violence | - | −0.631 ** | - |
| **Mental health problems:** | 0.131 | - | 0.484 |
| **Drug profile before prison:** |  |  |  |

**Table 4.** *Cont.*

|  | **TOTAL** | **MUDRES** | **REINAC** |
|---|---|---|---|
| AA | 3.776 *** | −0.827 | - |
| NA | −0.809 | −21.075 | - |
| EX | −1.800 | −0.905 | - |
| **Current drug profile:** |  |  |  |
| AA | −1.041 | −0.398 | - |
| NA | −0.331 | −0.938 ** | - |
| EX | 1.361 | 18.033 | - |
| **Criminal antecedents as a minor:** | 3.317 *** | - | 2.940 ** |
| **Age first went to prison:** |  |  |  |
| 18–25 years | −18.060 | - | 4.627 *** |
| 26–36 years | −20.410 | - | 2.999 *** |
| 37–49 years | −22.578 | - | 1.800 |
| 50 years or older | −26.887 | - | - |
| Persons close to respondent who are in prison | 0.080 | 0.951 *** | −0.201 |
| **Type of crime:** |  |  |  |
| Against the integrity of persons | −0.186 | −0.166 | 19.794 |
| Against property and the socio-economic order | 0.237 | 1.034 ** | 20.348 |
| Against collective safety | −0.213 | 0.434 | 20.101 |

*** Significant at 99%; ** significant at 95%; Source: The authors (Projects Ref. EDU2009-and Ref. EDU2016-79322-R).

First, if we analyze the specific model for the total sample, we see that the $p$-value < 0.05 for the variable age for age groups 18–25 years old, 26–35 ($p$-value = 0.000 in both cases), and 36–45 ($p$-value = 0.042). A relationship thus exists between this variable and whether or not a woman is a recidivist. Another related variable is national origin ($p$-value = 0.03), as Spanish women showed a greater likelihood of being recidivists than do foreign women. A relationship is also established between drug profile before prison and recidivism ($p$-value = 0.002). Specifically, being AA increased women's probability of recidivism, as does the fact of having criminal antecedents as a minor ($p$-value = 0.008).

Second, as in the previous model, the model developed for MUDRES showed a relationship between recidivism and national origin ($p$-value = 0.04 < 0.05). In this case, the probability of recidivism increases among Spanish women. Another related variable is gender violence ($p$-value = 0.032). The probability of recidivism decreases with gender violence. Drug profile at the time of the study also shows a significant relationship to decreasing probability of recidivism for NA ($p$-value = 0.021). Another variable with a $p$-value = 0.002 < 0.05 (thus establishing a relationship) is having persons close to one who are in prison. This condition increases the probability of recidivism, as does having committed a crime against property and the socioeconomic order ($p$-value = 0.035).

Third, in the model for REINAC, as in the two previous models, we observe a relationship between recidivism and national origin ($p$-value below 0.05), with the probability of recidivism increasing among Spanish women. Another significantly related variable is marital status. Being in a relationship decreases the probability of recidivism. A relationship is also established with whether or not one has a criminal record as a minor ($p$-value = 0.000). In this case, having a criminal record as a minor increased the possibility of recidivism. In the REINAC sample, we also see a relationship between recidivism and age the respondent first went to prison. This relationship is significant for the age groups 18–25 years old ($p$-value = 0.000), 27–49 ($p$-value = 0.005), and over 50 ($p$-value = 0.001).

### 3.6. Relationships of Variables of Protective Factors against Recidivism

For protective factors, as for risk factors, we first performed a descriptive analysis to determine whether or not a relationship existed among the variables, using the $\chi^2$ test. Due to the scarcity of significant relationships (in contrast to those found for risk factor variables), we were unable to propose any regression model.

As Table 5 shows, the $\chi^2$ tests indicate a relationship between recidivism and having received official education during one's sentence in all three samples—total, MUDRES, and REINAC, with *p*-value = 0.000, *p*-value = 0.020, and *p*-value = 0.004, respectively. In REINAC, the data also establish a relationship to the variable on attending courses and/or programs during one's sentence (*p*-value = 0.028).

**Table 5.** Significant results of $\chi^2$ test for variables corresponding to protective factors.

| Relationship of Recidivism to: | Total | MUDRES | REINAC |
|---|---|---|---|
| Person(s) with whom one lived before prison | 0.812 | 0.861 | 0.951 |
| Person(s) with whom one will live after the sentence | 0.332 | 0.208 | 0.932 |
| Employment status before going to prison | 0.908 | 0.672 | 0.407 |
| Courses and/or programs attended during the sentence | 0.147 | 0.990 | 0.028 ** |
| Official education during the sentence | 0.000 *** | 0.020 ** | 0.004 *** |
| Family support before prison | 0.804 | 0.917 | 0.881 |
| Support from friends before prison | 0.196 | 0.735 | 0.562 |
| Family support now | 0.707 | 0.290 | 0.483 |
| Support from friends now | 0.425 | 0.957 | 0.780 |

*** Significant at 99%; ** significant at 95%; Source: The authors (Projects Ref. EDU2009-13408 and Ref. EDU2016-79322-R).

Table 6 presents the percentage distribution of the relationship between recidivism and having or not having received official education during one's sentence. In the total sample, 66.7% (149) of recidivists had pursued official education during their sentence, and 31.3% (68) had not. For non-recidivists, the figures were 54.3% (290) and 45.7% (244), respectively. In MUDRES, 67.4% (95) of recidivists had received official education, and 32.6% (46) had not, as opposed to 55.7% (170) and 44.3% (135) of non-recidivist women, respectively. In REINAC, 71.1% (54) of women recidivists had received official education, as opposed to 28.9% (22) who had not. For non-recidivists, the percentages showed that 52.4% (120) had pursued official education, and 47.6% (109) had not.

**Table 6.** Relationship between recidivism and official education during the sentence.

| | TOTAL | | MUDRES | | REINAC | |
|---|---|---|---|---|---|---|
| | Recidivist | Non-Recidivist | Recidivist | Non-Recidivist | Recidivist | Non-Recidivist |
| **No** | 68 (31.3%) | 244 (45.7%) | 46 (32.6%) | 135 (44.3%) | 22 (28.9%) | 109 (47.6%) |
| **Yes** | 149 (68.7%) | 290 (54.3%) | 95 (67.4%) | 170 (55.7%) | 54 (71.1%) | 120 (52.4%) |

Source: The authors (Projects Ref. EDU2009-13408 and Ref. EDU2016-79322-R).

In the distribution of percentages for the relationship between recidivism and participation in courses and/or programs during one's sentence in the open environment (REINAC) (Table 7), neither the total nor the MUDRES sample established a relationship. We see that the majority of both recidivists and non-recidivists attended job-related courses

and training courses: 71.4% (55) of recidivists and 51.9% (120) of non-recidivists. Next comes attending training/socio-educational programs, with 18.2% (14) and 27.7% (64). The percentages for those who did not attend any program or course were 6.5% (5) for recidivists and 12.6% (29) for non-recidivists.

**Table 7.** Relationship between recidivism and participation in courses and/or programs during sentence in open environment (REINAC).

|  | RECIDIVIST | NON-RECIDIVIST | TOTAL |
|---|---|---|---|
| None | 5 (6.5%) | 29 (12.6%) | 34 (11%) |
| Job-oriented courses | 3 (3.9%) | 18 (12.6%) | 21 (6.8%) |
| Training/socio-educational programs | 14 (18.2) | 64 (27.7%) | 78 (25.3%) |
| Job-oriented and training programs | 55 (71.4%) | 120 (51.9%) | 175 (56.8%) |
| Total | 77 (100 %) | 231 (100%) | 308 (100%) |

Source: The authors (Project Ref. EDU2016-79322-R).

## 4. Discussion

The analyses performed in this examination of recidivism among women in Spain locate Spain at around 28.8% recidivists and 71.2% non-recidivists in the open environment (REINAC) and 31.6% in the closed environment (MUDRES). These differences are explained by the context, as the women who enter the open environment have more positive prison evolutionary characteristics than those in the closed environment, and their criminal profile is also usually lower than men's [14]. The differences agree with some estimates presented in the introduction [21,47]. The SGIP measures a general recidivism level of 31.6% [5], although the data do not distinguish either sex or context (closed or open). From the gender perspective, we also observe a lower probability of recidivism in women than in men [21,28,47–49]. In Catalonia, for example [28,50,51], recidivism among women is around 9.2%, as opposed to 15.4% among men, although our study confirms that these figures are higher.

For risk factors, the results show a relationship among the following variables:

1.　Age and recidivism: The average age for recidivists is 38.52 in the total sample, 36.41 in MUDRES, and 42.19 in REINAC. The difference in the statistics for open and closed environments is striking. In our study, this difference can be explained by the environment in which the women were serving their sentences, since women in the open environment have reached the last stage of their sentence and are thus older. We have no official data, however, on environment. According to data from SGIP [5] on the prison population in Spain, 77% of women inmates are 31–60 and their average age is 43. These figures agree with the Report on the Situation of Women Prisoners [52], which indicates that this population has been aging since 2006 and that the aging is not only due to the long periods of time in prison but must also be seen in the context of changes in the evolution of crime in Spain.

If we turn to the relationship between age and recidivism, the average age of women recidivists is 40.59. We find this relationship only in the total sample and the sample from the closed environment (MUDRES) (*p*-value = 0.000 in both cases). In the total sample, the age range with the most recidivism is 36–45 (48.8%). The same age group is the most common in the closed sample (MUDRES) (52.1%). The most frequent age range for non-recidivists is 26–45 (36.3%). Non-recidivist women are distributed more widely across the higher/lower age ranges. The logistical regression model for the total sample confirms

this distribution, showing a relationship between age and recidivism in the age groups 18–25, 26–35 (*p*-value = 0.000 in both cases), and 36–45 (*p*-value = 0.042), indicating that the probability of being a recidivist decreases as the women's age decreases. It is thus important when discussing criminality to attend to the individual's stage of life when committing criminal behavior, as crimes committed at a young age imply the development of future criminal behavior, as does having a criminal antecedent as a minor [53,54].

2. National origin and recidivism: In all cases, we see that being Spanish (as opposed to being foreign) increases the probability of becoming a recidivist, as 94% of recidivists were Spanish in the total sample, 92.9% in MUDRES, and 96% in REINAC. The sample of foreign women (25%) is, however, smaller than that of Spanish women (75%). Martín-Palomo and Miranda [55] help us to understand this finding on foreign women better. The data do not reflect crimes committed by immigrant residents of Spain. Rather, these women become foreigners at the same time as they become criminals. These foreign women are part of the "non-Spanish" population, including women without prior residence who enter the police, court, and Spanish prison system, where they are sentenced and serve sentences in Spanish prisons [56].

3. Education level and recidivism: The chi-square tests indicate a relationship to recidivism, *p*-value = 0.000 < 0.05, in all three samples. The education levels of recidivists are lower than those of non-recidivists. For recidivists, the most common education levels are no and primary education, and for non-recidivists secondary education and vocational training. We also find a higher percentage of women with more education among non-recidivists, but this relationship is not established in the regression model proposed.

Along these lines, other studies have shown that women inmates typically have low or medium-level education and follow cultural and educational paths with many disadvantages before going to prison [1,57], with 30% of women inmates in prison regimes having no education and 40% of women having only completed primary school [58]. Similarly, 66.6% of the young women left the education system as minors (primary 49.3% and secondary 22%), becoming more vulnerable than those who stayed in school to adult age [59].

4. Marital status, children, and recidivism: In REINAC, the sample also shows a relationship of recidivism to *marital status* and *children*, *p*-values = 0.003 and 0.037, respectively. It is worth noting, in the case of both recidivists and non-recidivists, that around 70% of the women do not have a domestic partner. They are thus more vulnerable and have greater need when maintaining self-worth and facing their family situation. Around 80–90% had children, the lower percentage referring to non-recidivist women. Most of the women in prison have family responsibilities, a factor with two types of consequences. One is negative, as the effects of going to prison extend to the rest of the nuclear family, especially to younger children. The other is positive, as having family responsibilities enables the women to focus on reintegration, since they preserve these bonds [60,61].

5. Gender violence and recidivism: In the MUDRES sample, we find a relationship between recidivism and gender violence (*p*-value = 0.034): 57.5% of recidivists were abused, while 42.5% were not. Among non-recidivists, 68% had pathologies and 32% did not. Most of the women had suffered gender violence. It is odd that the percentages are more polarized for non-recidivists and more similar for recidivists. This relationship is also confirmed in the model proposed (*p*-value = 0.032), which indicated that the probability of recidivism decreases with gender violence. It is important to stress that many authors relate women's criminality to histories of abuse and/or experience of gender violence [14,15,55,62–66]. Other authors identify the triggering factor as the degree to which the domestic partner urges the women to perform criminal or antisocial activities [65,67–69].

6. Mental health problems and recidivism: This relationship appears in both the total sample (*p*-value = 0.006) and REINAC (*p*-value = 0.013). In the total sample, 85.2% of recidivists suffered some self-perceived symptomatology of mental illness. The

percentage among non-recidivists was 76.6%. In REINAC, 86.8% of recidivists suffered some symptomatology, as did 72.9% of non-recidivists. We generally see that most of the women have some type of mental symptomatology and that this problem is more common among recidivists. The relationship was not significant in the regression model, however. In relation to the previous point (gender violence), some authors relate the fact of having experienced violence or abuse to mental health problems, which hinder interpersonal relationships and increase the probability of suicide and criminal behavior [70–72].

7. Addictions and recidivism: The variables on drug dependence profile include information from two different times, before prison and at the time of the study. We see a relationship of addiction profile (according to the classification used in the MUDRES project [14,15] (at both times) to whether one becomes a recidivist (*p*-value = 0.000) in both the total sample and in MUDRES. In both samples, over 50% of the women recidivists had had drug problems (AA, EX, and MMT). This percentage is lower for addiction profile at the time of the study. Non-recidivists, in contrast, include a higher percentage of women who had not had problems with drugs (NA). Añaños and García [14] find that 60.6% of the women's prison population had addiction problems before entering prison and that 24.7% continue to take drugs after completing their sentence.

Along the same lines, the regression model for MUDRES also established this relationship (*p*-value = 0.021) in the variable drug profile at the time of study, indicating that the probability of recidivism decreases if one is NA.

Numerous studies have analyzed the link between an increase in criminal activity and drug dependence [14,73–75] and find a greater incidence of crime in drug users who consume psychoactive substances.

We thus see that drug consumption not only motivates crime but also limits social reintegration due to its interference with social support and opportunities for finding work [34,37,40,43]. Mallik-Kane and Visher [37] indicate that women's probability of finding work or social support decreases in relation to substance abuse and diagnosis of mental illness. Such women would, in turn, be more likely to become recidivists [23]. This situation requires ongoing professional intervention and treatment.

8. Criminal antecedents as a minor and recidivism: A relationship is established in both the total sample and in REINAC, *p*-values = 0.019 and 0.000, respectively. We observe that the percentages of women with criminal antecedents as minors are somewhat higher for recidivists. The regression model proposed also identifies this relationship in the total sample (*p*-value = 0.008) and in REINAC (*p*-value = 0.000). Having criminal antecedents as a minor increases probability of recidivism in women [36,50,53,54].

9. Age of first prison sentence and recidivism: For this variable, a relationship is established in the total sample and in REINAC (*p*-value = 0.000) in both cases. It is interesting that women are more likely to become recidivists if they first went to prison at a younger age. For non-recidivists, the most common age group for the first prison sentence was 26–49 years old. The regression model for REINAC establishes a relationship for age groups of 18–25 (*p*-value = 0.000), 27–49 (*p*-value = 0.005), and 50 or over (*p*-value = 0.001). Thus, the earlier a woman first goes to prison, the greater the likelihood that she will become a recidivist. These results are concerning and demonstrate the need for earlier intervention. The specialized literature indicates that beginning criminal activity young directly conditions subsequent criminal processes [36,50,54]. However, this fact is not entirely reliable when indicating a greater possibility of recidivism [50,76].

10. Persons close to the respondent who are in prison and recidivism: A significant relationship is established in the total sample, MUDRES (*p*-value = 0.000 in both cases), and REINAC (*p*-value = 0.001). In the total sample, the percentage distribution for recidivists was 66% of women who had people close to them who were in prison and 34% who did not. In MUDRES, 68.3% of recidivists had someone close to them who was in prison, and 31.7% (44) did not. In REINAC, 61.8% had someone close

to them, and 38.7% did not. We confirm the highest percentage to be recidivists who have people close to them who are in prison. More non-recidivists, in contrast, do not have people close to them in prison. The regression model for MUDRES (*p*-value = 0.002) establishes this relationship, indicating that having people close to one who is in prison increases the probability of committing another crime. The same occurs in the case of the type of crime against property and the socioeconomic order. Other studies, such as those performed by Segeren, Fassaert, De Wit, and Popma [77], Scott and Brown [78], and Graña, Andreu, and Silva [21], consider this situation as one of the main risk factors of recidivism and of criminal behavior in general.

11. Type of crime and recidivism: As in previous cases, a relationship exists in the total sample, MUDRES (*p*-value = 0.000 in both cases), and REINAC (*p*-value = 0.001. In all three cases, the most common type of crime in recidivist women is against property and the socioeconomic order, and the least common is against the integrity of persons. In non-recidivists, however, the most common crime is against collective security. In the results obtained in the model tested, only the case of MUDRES, specifically for the case of committing a crime against property and the socioeconomic order (*p*-value = 0.035), shows a relationship; having committed this type of crime increases the probability of recidivism. The study by Van der Put, Assink, and Gubbels [79] showed that risk factors are generally more closely related to recidivism involving non-violent crimes than involving violent crimes, with only a weak relationship or none at all to sex crimes, requiring different focuses on the treatment of criminal recidivism.

For protective factors and recidivism, we find that:

The evidence found in the study shows the importance of the relationship between recidivism and having received no official education during the sentence. The $\chi^2$ tests indicate a relationship between recidivism and having received official education during the sentence in the total sample, MUDRES, and REINAC (*p*-value = 0.000, *p*-value = 0.020, and *p*-value = 0.004, respectively). In all three cases, the majority of the women received official education during their sentences, with the highest percentages for recidivists (66.7% total sample, 67.4% MUDRES, and 71.1% REINAC). This finding is due, as we saw before, to these women's low education level [1,58] and reveals the importance of official education in prison intervention for women—in this case, specifically in matters of recidivism prevention. Adhering to this policy is important from the socioeducational perspective, based on which goals are proposed and content-oriented to reintegration and prevention of recidivism.

Only in REINAC was a relationship established to the variable on attending courses and/or programs during one's sentence (*p*-value = 0.028). The majority of both recidivists and non-recidivists attended job-related courses and training/socioeducational programs. Only 6.5% of women recidivists and 12.6% of non-recidivists did not attend any program or course in the open/temporary release regime. As in the case of official education, this result is due to the women inmates' lack of education [14,17,42,80,81].

As to the relationship between formal education and attending training courses, Añaños et al. [59] show that training in prison can become an opportunity to increase prosocial attitudes, improve emotional wellbeing, and decrease the risk of unemployment. These opportunities become the starting point for preparing for life in freedom.

## 5. Conclusions

This study presents an in-depth examination and analysis of women's prison recidivism in Spain, from socioeducational, gender, and SDG perspectives, including endorsing inclusive, equitable, high-quality education, independently of the deprivation of freedom in which the women are living, as well as promoting opportunities for lifelong learning, fostering gender equality and empowerment of women, and promoting peaceful, inclusive societies for sustainable development, thereby facilitating social reintegration. These are issues of great importance in a population already inherently damaged and excluded [3].

These goals provide foundations for understanding the current state of this issue, social justice, professional practice, decision-making on criminological policies, and tasks such as the design of new prevention and intervention plans, as well as for calibrating instruments to evaluate risk and application of new measures for prison management [21].

We find a profile of women recidivists who are on average 40.58 years old, with the most common age range being 36–45. They are generally Spanish and have a low education level, principally primary or no education. In terms of marital status, they do not have a domestic partner (are single, separated/divorced, or widowed). The great majority of these women have children. A high percentage have also experienced some self-perceived symptomatology of mental illness, have had problems with addiction, and have persons close to them who are in prison. Their criminal profile typically includes criminal antecedents as a minor, first going to prison between the ages of 18–25, and most often having committed crimes against property and the socioeconomic order and against collective security. These data clearly place the women in a situation of greater vulnerability to commit a crime and to repeat criminal activity. It is important to indicate, however, that some of these characteristics change based on the prison regime in which the women are serving their sentences, the time during the sentence at which they engaged in the study, whether they were in the closed or open context, and to how their needs change based on prison intervention (e.g., the change undergone by persons with profiles of addiction).

The issues tackled here aim to optimize prison intervention and treatment to make it more sustainable, taking into account the special characteristics of gender, needs, interests, potentials, etc., of the population studied [8,18,73] in order to influence human development in prison despite the condition of isolation. Education and training play a crucial role in this respect, influencing favorable social reintegration processes. On the one hand, in following the prevention strategies proposed by UNODC [2], our proposal can contribute to sustainable development, focusing on the reduction of violence, fostering social inclusion, promoting reintegration, and strengthening victims. On the other, it can encourage autonomy, self-concept, decision-making, empowerment, management of everyday life, and assumption of responsibility for one's situation and change processes, among other effects.

Finally, integration into both life in prison and processes of socioeducational action (among others) seek to dignify the person in preparing her for freedom, developing various competencies and using all available media and resources for this purpose [34,82], while also strengthening the training of professionals through greater efficiency in the availability of support resources through education [62], both formal and socioeducational. Educational action is thus a vital tool for re-education and social reintegration (personal, social, work-related, etc.), as it develops community preventive functions and generates new approaches through elements and actions for socialization and prevention of recidivism. All of these recommendations adopt approaches to sustainable human development in the prison context, following the principle of increasing the wellbeing of persons and their communities.

**Author Contributions:** Conceptualization, F.T.A. and E.M.-L.; methodology, F.T.A. and E.M.-L.; software, E.M.-L.; formal analysis, E.M.-L.; acquisition of funding, F.T.A.; research, F.T.A. and E.M.-L.; project administration, F.T.A.; Resources, F.T.A.; Supervision, F.T.A.; Validation, F.T.A. and E.M.-L.; Visualization, F.T.A. and E.M.-L.; writing: first draft, E.M.-L.; writing: revision and editing, F.T.A. and E.M.-L. All authors have read and agreed to the published version of the manuscript.

**Funding:** This paper was prepared within the framework of the R&D&I Research Project REINAC-"Reintegration and support processes for women on temporary release", Reference EDU2016-79322-R (2016–2020), financed by the Spanish National Plan for Research, R&D&I Research Projects, the Spanish Ministry of Economy and Competitiveness (MINECO), the State Research Agency (AEI), and FEDER, Spain. PI: Fanny Tania Añaños Bedriñana.

**Institutional Review Board Statement:** Not applicable.

**Informed Consent Statement:** Applicable. In it, the rights of anonymity and voluntary participation in the research were explained to the research participants.

**Data Availability Statement:** The data presented in this study are available on request from the corresponding author and the R & D & I Research Projects of the Ministry of Economy and Competitiveness of Spain (MINECO).

**Conflicts of Interest:** The authors declare that they have no conflict of interest.

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
