# Peer review of "Factors of Prison Recidivism in Women: A Socioeducational and Sustainable Development Analysis"

_sustainability, doi:10.3390/su13115822_

Round 1

Reviewer 1 Report

This paper is interested in analysing the risk and protective factors that are closely related to the recidivism of women sentenced to prison in Spain. For this purpose, a quantitative study is carried out using the survey method with an impressive and representative sample. The findings highlight the relationship between recidivism and sociological differences and criminological characteristics. The nationality of the inmates is presented as a risk factor for the group of inmates in a closed and in an open environment. In relation to the protective factors, the studies undertaken in their life trajectory and those completed while in prison are included. The research evidence corroborates the interest of the article for the educational community in particular and for society in general. Education in prisons and the reintegration of prisoners, especially women, should be a priority issue and the scarcity of studies on this subject makes this work a highly relevant manuscript.

In general terms, I consider that the subject of the manuscript is very interesting for the journal Sustainability because of its social impact but also because of its potential scientific impact. In addition, the authors do a good job of reviewing the state of the art by consulting a large number of scientific papers; although it is recommended that the review of scientific papers from the last three years be extended. This could improve the introduction of the article and, of course, the discussion.  

In relation to the methodological section, the article is right to carry out a study with a large number of participants. It is also representative of the population under study. It is recommended that the authors include the project data in a Funding section at the end of the manuscript. 
On the other hand, the instruments used are appropriate for the research. However, it would be desirable to indicate some consideration of the reliability of the quantitative part of the instrument and some assessment of the construct validity of the survey (how was the questionnaire created? was there any kind of content validation by external judges?) In relation to the statistical tests, the analyses seem adequate. However, in the results it is recommended to follow APA and to adjust decimals with full stops instead of commas. Finally, the discussion and conclusions of the study are really good although adding some references from the last three years could be improved. 

Author Response

Response to Reviewer 1’s Comments

Dear Reviewers,

We wanted to thank you for the suggestions and indications you have given us for this version of the manuscript, as you have enriched and improved it.

Below we detail and explain the issues raised in the review of the article "Factors of prison recidivism in women: A socio-educational and sustainable development analysis".

We comply with and send corrections according to suggestions made.

We trust it to be to your satisfaction.

Waiting for your answer.

A cordial greeting,

The authors

(Elisabet Moles and Fanny T. Añaños).

Point 1: It is recommended that the review of scientific papers from the last three years be extended. This could improve the introduction of the article and, of course, the discussion. And, the discussion and conclusions of the study are really good although adding some references from the last three years could be improved.

Response 1:  Following the reviewer’s recommendations, we have expanded the review of scholarly articles from 2017 to the present in the Introduction, Discussion, and Conclusions sections, taking into account the scarcity of information on the topic analyzed in the article (in red). (Lines 594-595, 601-602, 626-627, 628-629, 666-667, 676-677, 781-782)

Point 2: It is recommended that the authors include the project data in a Funding section at the end of the manuscript.

Response 2: Following the reviewer’s recommendations, we have included the Project data in a Funding section in the manuscript (in red). (Lines 584-586)

Point 3: It would be desirable to indicate some consideration of the reliability of the quantitative part of the instrument and some assessment of the construct validity of the survey (how was the questionnaire created? was there any kind of content validation by external judges?)

Response 3: As the reviewer suggests, we have added more information on the instruments used in the study in order to clarify the validity of the study questionnaire (in red). (Lines 164-165)

Point 4: In relation to the statistical tests, the analyses seem adequate. However, in the results it is recommended to follow APA and to adjust decimals with full stops instead of commas.

Response 4: Following APA format and the reviewer’s instructions, we have changed the decimals from commas to periods (in red). (Throughout the manuscript)

Reviewer 2 Report

Dear Authors

 The study you presented in this paper brings a fresh perspective to the issue of prison recidivism by introducing a gender perspective, which is not usual. Therefore it brings a more focused insight on both the risk and protective factors for women recidivism in Spain. It may have interest for the journal`s readers since it addresses social sustainability, but some major revisions of key elements should be addressed.

Its results are drawn from two different projects coordinated by Fanny T. Añanos Bedriñana. This raises the first concern, since both projects were carried out in different time frames (2010-2013, the first and 2017-2020 the second). Also, they apparently varied in terms of coverage (the first covered 42 prisons in 11 autonomous communities and the second covered 31 prisons in 13 autonomous communities).

The second study, although brings forward more UpToDate data raises another set of concerns since it addresses women on temporary release. It would be imperative to describe the criteria according to which women are selected for this program. This criterion may have a huge impact on the results concerning recidivism. Therefore this information should be presented in the article before comparing data from the two projects. Take addiction profile as an example. Is it taking into consideration when deciding the integration of inmates in an open regime? If so how does it affect the results?

Furthermore, a more rigorous characterization of the sample should be presented. If we don’t know, for example, what was the total percentage of Spanish women in the studies we cannot establish a link between nationality and recidivism. The only data disclosed is age. Why aren`t the other data presented?

Focusing on age group, the article finds that most recidivists women era on the 36-45 years old range, on both closed and open environment. However, the article fails to show the relevance of these results, when we take into consideration the average age of these population 36,41 for the closed environment and 42,19 in the open regime.

It would also be important to compare results with non-recidivists, which doesn`t occur systematically. For example, is the percentage of Spanish women also higher among non-recidivists?

In some cases, this lack of information weakens the overall conclusions presented. If one of the major conclusions is that formal and socio-education is a vital tool for reintegration, isn`t it important to know the percentage of non-recidivists according to education level? The authors refer in lines 562-563 that the majority of both recidivists and non-recidivists attended job-related courses and socio-educational programs. Isn`t that a contradiction?

These issues have a negative impact on the scientific soundness and the overall merit of the article since they weaken the results and their discussion.

The quality of the presentation should also be improved, the data are shown in tables and presented in text, further along, the article the discussion section summarizes, once more, these results. This spiral structure can be repetitive and confusing in some parts. The authors should consider changing the structure of the article and maybe combine results and discussion.

Some specific parts should also be reviewed:

Lines 25/26: Sustainable Development Goals (SDG`s) is presented as one of the Keywords, but, although mentioned in the introduction, the paper doesn’t really discuss the results of the study in relation to these goals. The same can be said for the conclusion, where, once more the SDG`s are mentioned without really discussing the repercussion of this study to the purpose of these goals. We suggest that this keyword be removed or replaced.

Lines 464/465: review the numbers presented “57.5% of recidivists were abused, while 4.5% were not”

The English should be reviewed and small rectifications should be done. See for example:

Line 57: revise the expression “lower than of men”.

Lines 58/59: check the verb tense in the sentence.

Line 68: check the verb tense “performe if recidivism” 

Line 92: revise the expression “in the prison environment”.

Best Regards

Author Response

Dear Reviewers,

We wanted to thank you for the suggestions and indications you have given us for this version of the manuscript, as you enriched and improved it.

The authors,

Elisabet Moles and Fanny T. Añaños 

Round 2

Reviewer 2 Report

Dear Authors

After careful review of the new version of the manuscript concerning women`s prison recidivism in Spain, we find that the changes introduced are sufficient to improve the overall merit of the paper.

Although we still consider that it would benefit from a brief clarification of the criteria used in accepting women in a temporary release program in Spain. We still have some concern about the impact of these criteria in the results presented.

Best Regards

Author Response

Dear Reviewer,

First of all, thank you for the suggestions and corrections you have made to us in order to improve the quality of this version of the text, which brings more rigor, if possible, to the work presented.

We send corrections according to the nuances made.

We trust it to be to your satisfaction.

Waiting for your answer.

A cordial greeting,

Authors: Elisabet Moles-López and Fanny T. Añaños 
